

# HLA alleles measured from COVID-19 patient transcriptomes reveal associations with disease prognosis in a New York cohort

René L. Warren[1] and Inanc Birol[1]

Genome Sciences Centre, BC Cancer, Vancouver, CA-BC, Canada

## ABSTRACT

**Background**. The Human Leukocyte Antigen (HLA) gene locus plays a fundamental role in human immunity, and it is established that certain HLA alleles are disease determinants. Previously, we have identified prevalent HLA class I and class II alleles, including DPA1*02:02, in two small patient cohorts at the COVID-19 pandemic onset.
**Methods**. We have since analyzed a larger public patient cohort data ($n = 126$ patients) with controls, associated demographic and clinical data. By combining the predictive power of multiple *in silico* HLA predictors, we report on HLA-I and HLA-II alleles, along with their associated risk significance.
**Results**. We observe HLA-II DPA1*02:02 at a higher frequency in the COVID-19 positive cohort (29%) when compared to the COVID-negative control group (Fisher's exact test [FET] $p = 0.0174$). Having this allele, however, does not appear to put this cohort's patients at an increased risk of hospitalization. Inspection of COVID-19 disease severity outcomes, including admission to intensive care, reveal nominally significant risk associations with A*11:01 (FET $p = 0.0078$) and C*04:01 (FET $p = 0.0087$). The association with severe disease outcome is especially evident for patients with C*04:01, where disease prognosis measured by mechanical ventilation-free days was statistically significant after multiple hypothesis correction (Bonferroni $p = 0.0323$). While prevalence of some of these alleles falls below statistical significance after Bonferroni correction, COVID-19 patients with HLA-I C*04:01 tend to fare worse overall. This HLA allele may hold potential clinical value.

## INTRODUCTION

Modern history has been plagued by deadly outbreaks, from the recurring influenza (*e.g.*, Spanish, Asian, Hong Kong, Avian) and HIV/AIDS viral pandemics, to bacterial and protist infections causing tuberculosis and malaria. Since the early 2000s, we have faced another threat: novel coronavirus infections causing severe respiratory illnesses such as SARS, MERS and today, coronavirus disease 2019—COVID-19 (*Petersen et al., 2020*). The SARS-CoV-2 coronavirus responsible for the COVID-19 respiratory disease is of particular concern; not only does SARS-CoV-2 spread quickly, the symptoms of its infection, when

Corresponding author
René L. Warren, rwarren@bcgsc.ca

exhibited, are very similar to that of the cold and flu making it difficult to diagnose, trace and contain. Further, infections by this virus affects different individuals differently. For instance, older men ($\geq$65 years old) with pre-existing medical conditions, such as diabetes, appear at increased risk of progressing into the more severe phase of the disease, yet SARS-CoV-2 infections affect all other age groups evenly except occasionally the children and adolescents (*Guan et al., 2020*). Most peculiar is that a high proportion of individuals who tested positive for SARS-CoV-2 are asymptomatic—as high as 43% recorded in Iceland, a rate that appears to vary depending on jurisdictions and populations (*Gudbjartsson et al., 2020*; *Mizumoto et al., 2020*; *Nishiura et al., 2020*). As of now, the disparity in patient response to SARS-CoV-2 infection is still eluding us.

The most efficient way to combat pathogens has been through the use of our own defense mechanism: our acquired immunity. This is done by vaccination campaigns that effectively prime our immune systems at the population level before we even encounter pathogens. But design of effective vaccines must consider interactions with host immune genes. The Human Leukocyte Antigens (HLA) are a group of such genes encoding surface receptors that bind short peptide epitopes derived from endogenous (class I) or exogenous (class II) antigens, including viral antigens, and they facilitate killer or helper T cells to set off an appropriate immune response. The magnitude of this response varies between patients as populations and individuals have different composition of HLA genes and variable T cell repertoires. As such, HLA induces a bias, which is responsible for documented host susceptibility to disease (*Dendrou et al., 2018*). Some of the notable associations between HLA and disease are observed in AIDS patients, with certain HLA alleles conferring protection (*Goulder & Walker, 2012*). In other cases, HLA has been implicated with autoimmune diseases and diabetes (*Chu et al., 2018*; *Noble et al., 2002*; *Ollila et al., 2015*; *Tafti et al., 2016*). The exact underlying mechanisms behind these associations are unclear, but there is mounting evidence that bacterial and viral infection may be the trigger for some (*Ollila et al., 2015*) and that HLA plays a critical role in the viral infection cycle, including viral entry into host cells (*Karakus et al., 2019*).

Since the beginning of the pandemic, worldwide reports have emerged on host susceptibility to COVID-19 (*Ovsyannikova et al., 2020*; *COVID-19 Host Genetics Initiative, 2021*), including reports of possible associations with HLA (*Correale et al., 2020*; *Novelli et al., 2020*; *Pisanti et al., 2020*; *Poulton et al., 2020*; *Khor et al., 2021*; *Wang et al., 2019*; *Warren & Birol, 2020a*; *Weiner et al., 2020*; *Langton et al., 2021*; *Augusto et al., 2021*) and reports with absence of a definite link in certain populations (*Severe Covid-19 GWAS Group et al., 2020*; *Ben Shachar et al., 2021*). Using publicly available metatranscriptomic sequencing data made available at the pandemic onset, we had demonstrated the utility of a high throughput *in silico* method for characterizing the HLA types of COVID-19 patients from bronchoalveolar lavage fluid and blood samples and reported on prevalent alleles, including the DPA1*02:02P - DPB1*05:01P HLA-II haplotype observed in seven out of eight of patients from two small cohorts (*Warren & Birol, 2020a*). Our early report had demonstrated the practical utility of HLA prediction on shotgun sequencing data derived from clinical COVID-19 host samples. Here, using public RNA-seq sequencing data from a larger patient cohort of New York patients with ($n = 100$) and without ($n = 26$) COVID-19

and with clinical outcomes and demographics data (*Overmyer et al., 2020*), we report on HLA alleles associated with disease susceptibility (DPA1*02:02), and severity (A*11:01, C*04:01) and present our findings in light of available demographic characteristics using hospitalization and disease severity metrics.

## MATERIALS & METHODS

We downloaded Illumina NOVASEQ-6000 paired-end (50 bp) RNA-Seq reads from libraries prepared from the blood samples of 126 hospitalized New York, NY (USA) patients, with ($n = 100$) or without COVID-19 ($n = 26$) (ENA project: PRJNA660067, accessions: SRX9033799–SRX9033924). This data is part of a large-scale multi omics study from the Department of Molecular and Cellular Physiology, Albany Medical College, Albany, NY, USA, with aims to analyze COVID-19 Severity. Clinical as well as demographics data was made available by the study authors (*Overmyer et al., 2020*) (GEO accession GSE157103, Table S1). As per their study, "Patients were considered for enrollment if they were older than 18 years and were admitted to the hospital due to symptoms compatible with COVID-19 infection" and "Patients who tested positive for COVID-19 were later assigned to that specific group and analyzed accordingly, and the COVID-19 negative group was composed of the remaining individuals." In their study, patients were tested for COVID-19 by nasopharyngeal swab kit testing using the Abbott Realtime SARS-CoV-2 Assay® (Abbott, IL), as described in a follow-up study that used the same data (*Balnis et al., 2021*). We first evaluated the data for consistency (Fig. S1). On each patient RNA-Seq dataset, we ran HLA prediction software OptiType (*Szolek et al., 2014*) (v1.3.4), seq2HLA (*Boegel et al., 2012*) (v2.3), and HLAminer (*Warren et al., 2012*) (v1.4 targeted assembly mode with defaults) as described (*Warren & Birol, 2020a*). We tallied HLA class I (HLA-I) and class II (HLA-II, supported by Seq2HLA and HLAminer only) allele predictions and for each patient we report the most likely HLA allele (4-digit resolution), indicating HLA predictor tool support (Tables S2 and S3). For HLA-I predictions, we opted to report the OptiType base predictions with additional support from seq2HLA and HLAminer, when available, due to the former tool's higher predictive performance. Also, because we only had predictions from two tools for HLA-II alleles and the predictions were generally less reliable, we based the predictions on Seq2HLA, tracking additional support from HLAminer when available. HLA-II alleles with low occurring frequency in the COVID-19 cohort and with no secondary predictions from another tool (*e.g.*, HLA-II DPB1*105:01) were omitted from further analysis.

Looking at class I and II alleles predicted in 10% or more of COVID-19 positive patients (class I, $n = 17$; class II, $n = 25$) we calculated Fisher's Exact Test (FET), first testing for enrichment in COVID-19 positive *vs.* negative patients (R function fisher.test, alternative = "greater"). We filtered out HLA alleles occurring in less than 10% of COVID-19 patients to reduce confounding effects due to the small cohort size, while keeping HLA alleles occurring at frequencies expected in heterogeneous populations. For those same alleles (found in ≥10% patients) and inspecting only the COVID-19 positive cohort, we tested for the probability of patient hospitalization, as measured by the Intensive Care Unit (ICU)

admittance reported by the original study authors, using FET. We looked further into the risk of hospitalization in COVID-19 patients with *vs.* without these alleles using the Kaplan–Meier (KM) estimator (R library survival), plotting the probability of remission using the "hospital-free days post 45 day followup (days)" (HFD-45) metric reported by the study author as a proxy for disease severity, with lower HFD-45 numbers indicating worse outcomes. Similarly, we ran the KM estimator using another metric of disease severity, "ventilator-free days", which captures the most severe cases with COVID-19 patients suffering respiratory deterioration and requiring mechanical ventilation. On each set we calculated the log-rank $p$-value (R library survminer) and corrected for multiple hypothesis testing (Bonferroni correction) using the number and patient abundance rank of class I ($n = 17$) or class II ($n = 25$) HLA alleles observed in 10% or more of COVID-19 patients. As we predict HLA alleles from each patient, we tally the number of patients with a given allele and rank the allele by decreasing abundance of patient counts for each allele present in at least 10 COVID-19 patients (*i.e.,* 10% in a cohort of 100 patients) and used those values as the number of hypotheses for further Bonferroni correction of the FET $p$-values reported herein. We also inspected the combined influence of HLA alleles and patient demographics data (age, sex, ethnicity) on the hospitalization (ICU negative *vs.* positive) outcomes of COVID-19 patients, using odds-ratio calculations (R function fisher.test, and applying Haldane correction (*Valenzuela, 1993*) on zero values, when necessary). HLA haplotype association with disease severity was also tested. In all, 861 haplotypes were assessed for higher risk of hospitalization in the Overmyer cohort using the KM estimator, as described above, but none showed significant associations (data not shown). The statistical test employed in our study (Fisher's Exact Tests and log-rank for survival analysis, with Bonferroni corrections based on reported alleles above a minimum threshold) were principled and the underlying data based on fair observations and assumptions. We acknowledge that this study is underpowered due to the small cohort size.

## RESULTS

We collated the HLA class I and class II predictions of three *in silico* HLA predictors derived from the RNA-seq samples of a recent (*Overmyer et al., 2020*) COVID-19 positive patient cohort ($n = 100$) with control patients ($n = 26$) who tested negative for COVID-19 (Tables S1–S3). Due to the limiting short read length (paired 50 bp) we chose to first report on OptiType (*Szolek et al., 2014*) and seq2HLA (*Boegel et al., 2012*) class I and class II predictions, and count the additional allele support from seq2HLA and HLAminer (*Warren et al., 2012*). In all, we identify 17 and 25 HLA class I and class II alleles predicted in 10% or more of COVID-19 patients, respectively (Tables 1 and 2). There were many more alleles predicted (133 and 116 distinct HLA class I and II, respectively), but too few patients are represented at lower cut-offs to compute meaningful statistics. First, we looked at the statistical enrichment (Fisher's Exact Test - FET) of each allele in the COVID-19 positive set, compared to the COVID-19 negative control group. We find HLA-I A*30:02 and HLA-II DPA1*02:02 allele enrichment nominally significant (FET $p = 0.0417$ and $p = 0.0174$) at

**Table 1 HLA-I alleles identified in 10% or more COVID-19 positive patients and statistical tests of enrichment in the *Overmyer et al. (2020)* COVID-19 positive (*vs.* negative) cohort and association with hospitalization.** Font in bold face indicates significant associations (Fisher's Exact Test) not corrected for multiple hypothesis tests.

| HLA-I | COVID+ patients | ICU- patients | ICU+ patients | COVID- patients | H1: Enriched in COVID+ Fisher's Exact Test *p*-value | Bonferroni correction n= abundance rank[*] | H1: Increased risk of hospitalization Fisher's Exact Test *p*-value | Bonferroni correction n= abundance rank[*] |
|---|---|---|---|---|---|---|---|---|
| A*02:01 | 30 | 15 | 15 | 12 | 0.9614 | 0.9614 | 0.5862 | 0.5862 |
| A*24:02 | 23 | 12 | 11 | 6 | 0.6161 | 1 | 0.5000 | 1 |
| C*07:02 | 22 | 13 | 9 | 7 | 0.7890 | 1 | 0.8865 | 1 |
| C*07:01 | 18 | 12 | 6 | 7 | 0.8991 | 1 | 0.9668 | 1 |
| **C*04:01** | 18 | 4 | 14 | 3 | 0.3231 | 1 | **0.0087** | **0.0435** |
| C*06:02 | 16 | 8 | 8 | 6 | 0.8708 | 1 | 0.6071 | 1 |
| B*51:01 | 16 | 10 | 6 | 2 | 0.2292 | 1 | 0.9143 | 1 |
| C*03:04 | 14 | 10 | 4 | 3 | 0.5171 | 1 | 0.9796 | 1 |
| C*15:02 | 14 | 8 | 6 | 1 | 0.1363 | 1 | 0.8060 | 1 |
| A*01:01 | 13 | 8 | 5 | 7 | 0.9742 | 1 | 0.8832 | 1 |
| A*03:01 | 13 | 6 | 7 | 5 | 0.8680 | 1 | 0.5000 | 1 |
| **A*30:02** | 13 | 6 | 7 | 0 | **0.0417** | 0.5004 | 0.5000 | 1 |
| B*07:02 | 12 | 7 | 5 | 5 | 0.8969 | 1 | 0.8217 | 1 |
| A*30:01 | 12 | 4 | 8 | 1 | 0.2016 | 1 | 0.1783 | 1 |
| B*44:02 | 11 | 7 | 4 | 4 | 0.8320 | 1 | 0.9001 | 1 |
| A*68:01 | 11 | 6 | 5 | 0 | 0.0697 | 1 | 0.7377 | 1 |
| **A*11:01** | 10 | 1 | 9 | 2 | 0.5320 | 1 | **0.0078** | 0.1326 |

**Notes.**
 [*]Set to 1 when the corrected values exceed that probability.

the $\alpha = 0.05$ level (Tables 1 and 2). However, when Bonferroni correction is applied for the number of HLA class I allele tests or when the abundance rank is factored in for A*30:02, the test is not significant (#alleles = 17, Bonferroni $p = 0.7089$; allele abundance rank = 12, Bonferroni $p = 0.5004$, respectively). For HLA-II DPA1*02:02, Bonferroni correction finds the test insignificant at the $\alpha = 0.05$ level for the number of hypothesis and the allele abundance rank (#alleles = 25, Bonferroni $p = 0.4350$; allele abundance rank = 4, Bonferroni $p = 0.0696$).

COVID-19 positive patients could be further stratified into those who were hospitalized and admitted to the Intensive Care Unit (Tables 1 and 2, ICU+), and those who were not (Table 1 and 2, ICU-). When computing FET statistics, we find HLA-I A*11:01 and C*04:01 significant at the $\alpha = 0.05$ level (Table 1; $p = 0.0078$ and $p = 0.0087$, respectively) but none remain significant after Bonferroni correction. Of note, of the 14 COVID-19 patients admitted to ICU who have the HLA-I C*04:01 allele, nine have a Charlson comorbidity index (*Charlson et al., 1987*) score of 3 or more (Tables S1 and S2). The Overmyer study authors (*Overmyer et al., 2020*) reported important disease severity metrics (HFD-45 and days without needing mechanical ventilation), which we used to assess the remission probability of COVID-19 patients having a specific allele using Kaplan–Meier estimation. We find patients of the Overmyer cohort with either A*11:01 (Fig. 1A) or C*04:01 (Fig. 1B)

**Table 2  HLA-II alleles identified in 10% or more COVID-19 positive patients and statistical tests of enrichment in the *Overmyer et al. (2020)* COVID-19 positive (*vs.* negative) cohort and association with hospitalization.** Font in bold face indicates significant associations (Fisher's Exact Test) not corrected for multiple hypothesis tests.

| HLA-II | COVID+ patients | ICU- patients | ICU+ patients | COVID- patients | H1: Enriched in COVID+ Fisher's Exact Test p-value | Bonferroni correction n= abundance rank* | H1: Increased risk of hospitalization Fisher's Exact Test p-value | Bonferroni correction n= abundance rank* |
|---|---|---|---|---|---|---|---|---|
| DQA1*01:02 | 52 | 23 | 29 | 13 | 0.5147 | 0.5147 | 0.1585 | 0.1585 |
| DPA1*01:03 | 46 | 25 | 21 | 17 | 0.9769 | 1 | 0.8421 | 1 |
| DQA1*03:02 | 35 | 19 | 16 | 7 | 0.2971 | 0.8913 | 0.7991 | 1 |
| **DPA1*02:02** | 29 | 13 | 16 | 2 | **0.0174** | 0.0696 | 0.3299 | 1 |
| DPB1*01:01 | 25 | 12 | 13 | 4 | 0.3701 | 1 | 0.5000 | 1 |
| DPB1*04:01 | 25 | 14 | 11 | 12 | 0.9959 | 1 | 0.8221 | 1 |
| DPA1*02:01 | 22 | 11 | 11 | 6 | 0.6578 | 1 | 0.5952 | 1 |
| DPB1*02:01 | 21 | 12 | 9 | 3 | 0.2117 | 1 | 0.8369 | 1 |
| DQA1*05:02 | 20 | 10 | 10 | 3 | 0.2453 | 1 | 0.5984 | 1 |
| DQA1*02:01 | 15 | 7 | 8 | 6 | 0.8967 | 1 | 0.5000 | 1 |
| DRB1*07:01 | 15 | 7 | 8 | 3 | 0.8967 | 1 | 0.5000 | 1 |
| DQB1*02:01 | 14 | 8 | 6 | 6 | 0.9192 | 1 | 0.8060 | 1 |
| DRB1*15:01 | 14 | 8 | 6 | 3 | 0.7008 | 1 | 0.8060 | 1 |
| DQB1*03:09 | 13 | 8 | 5 | 1 | 0.1662 | 1 | 0.8832 | 1 |
| DPB1*104:01 | 13 | 9 | 4 | 6 | 0.9385 | 1 | 0.9643 | 1 |
| DQB1*05:01 | 12 | 6 | 6 | 4 | 0.7913 | 1 | 0.6202 | 1 |
| DQB1*06:11 | 11 | 5 | 6 | 3 | 0.6811 | 1 | 0.5000 | 1 |
| DPB1*105:01 | 11 | 1 | 10 | 2 | 0.4712 | 1 | 0.0039 | 0.0702 |
| DRB1*15:02 | 11 | 5 | 6 | 0 | 0.1804 | 1 | 0.5000 | 1 |
| DQB1*06:02 | 11 | 5 | 6 | 3 | 0.6811 | 1 | 0.5000 | 1 |
| DRB1*13:01 | 11 | 4 | 7 | 4 | 0.8320 | 1 | 0.2623 | 1 |
| DQA1*05:01 | 10 | 5 | 5 | 4 | 0.8689 | 1 | 0.6297 | 1 |
| DPB1*05:01 | 10 | 5 | 5 | 1 | 0.2912 | 1 | 0.6297 | 1 |
| DRB1*11:01 | 10 | 4 | 6 | 5 | 0.9427 | 1 | 0.3703 | 1 |
| DPB1*04:02 | 10 | 8 | 2 | 0 | 0.2200 | 1 | 0.9922 | 1 |

**Notes.**
  *Set to 1 when the corrected values exceed that probability.

to be at a significant increased risk of hospitalization (log-rank $p = 0.0099$ and $p = 0.0082$, respectively). When applying multiple test corrections to account for allele abundance rank, only C*04:01 ($n = 5$, Bonferroni $p = 0.0410$) remained significant at the $\alpha = 0.05$ level. When looking at patients needing mechanical ventilators, a severe outcome in COVID-19 disease progression, we only find patients with C*04:01 to be at a statistically significant increased risk (Fig. 1C, log-rank $p = 0.0019$). Multiple hypothesis test correction retains the statistical significance of this allele when factoring both the number of HLA-I alleles tested ($n = 17$, Bonferroni $p = 0.0323$) and C*04:01 abundance rank ($n = 5$, Bonferroni $p = 0.0095$).

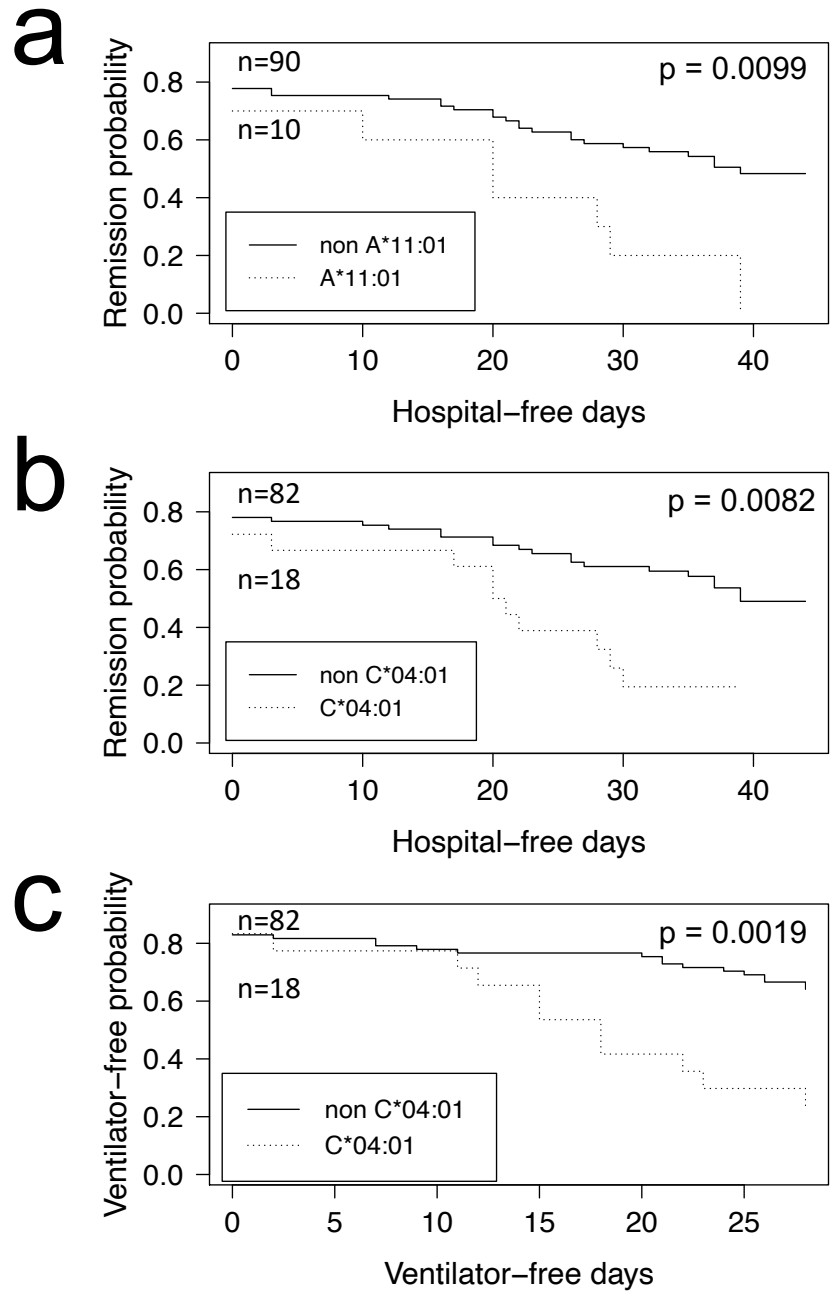

**Figure 1  HLA alleles associated with higher risk of hospitalization in a COVID-19 positive patient cohort.** COVID-19 positive patients were split into two groups per allele tested, depending on whether they were predicted to have the HLA allele under scrutiny or not. We ran the Kaplan–Meier estimator (R package survival) using the *Overmyer et al. (2020)* cohort HFD-45 metric for estimating the remission probability of patients without or with alleles (A) A*11:01 or (B) C*04:01 and mechanical ventilator-free days to estimate the statistical significance of the more severe disease outcome observed in COVID-19 patients with (C) C*04:01. Log-rank *p* values were calculated for each (R package survminer) and are indicated on the plots.

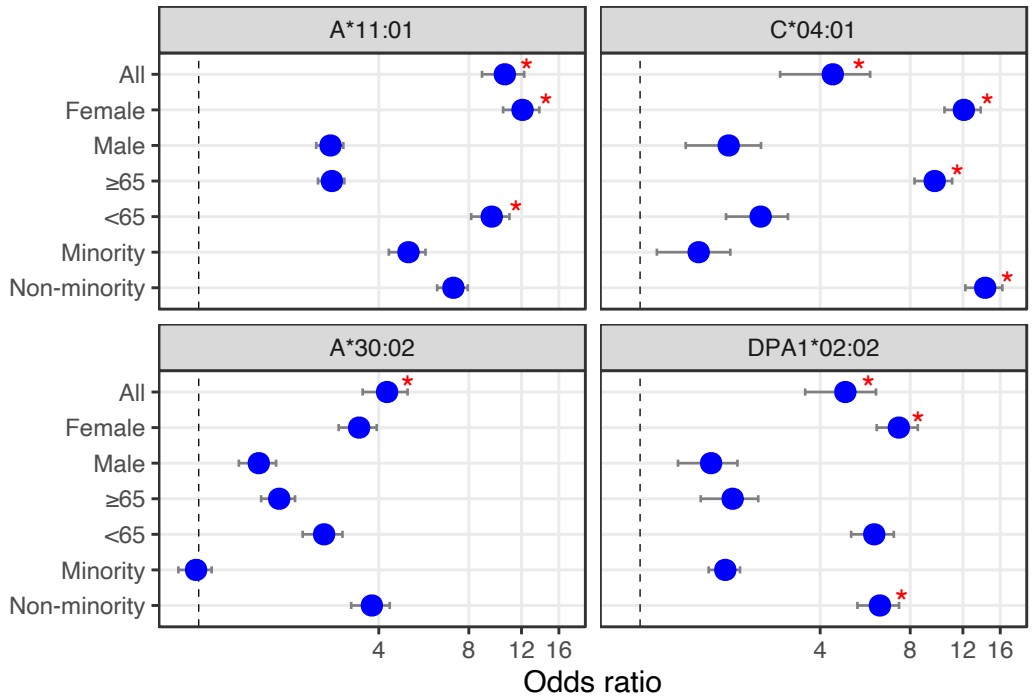

**Figure 2  HLA alleles – demographics combinations with analytic (bottom) or prognostic (top) potential in a COVID-19 cohort.** We calculated the odds ratio (OR) for each HLA-I and HLA-II allele observed in 10% or more of patients, and plotted OR and the influence of demographics for HLA alleles showing significant associations (from Tables 1 and 2). First, looking at the influence of demographic characteristics such as sex (Female/Male), age (65 years old or above/less than 65 years old) and ethnicity (Minority/Non-minority (*i.e.*, European ancestry)) on the susceptibility of patients with these alleles to test positive for COVID-19 (lower two panels), and on the risk associated with ICU hospitalization (upper two panels). Red asterisks indicate significant demographic characteristics at the alpha = 0.05 level (Fisher's Exact Test) not corrected for multiple hypothesis tests.

Looking at the influence of the aforementioned alleles in combination with simple demographics (sex, age and ethnicity), we find that of the Overmyer cohort patients with the DPA1*02:02 allele, those from a non-minority background and females appear at an increased risk of testing positive for COVID-19 in that cohort (Fig. 2 and, Table S4; odds ratio [OR] = 6.33 [5.33–7.34], FET $p = 0.0491$ and OR = 7.33 [6.18−8.48], FET $p = 0.0326$, respectively, where the numbers in square brackets represent the 95% confidence interval). The association with gender is also observed in alleles A*11:01 and C*04:01, putting female COVID-19 patients of this cohort at an increased risk of hospitalization for the class I alleles (Fig. 2 and Table S4; OR = 12.09 [10.41–13.76], FET $p = 0.0105$ for both). In this cohort, we also find patients with A*11:01 in the younger age group (<65 years old) at increased risk of hospitalization (Fig. 2 and Table S4; OR = 9.54 [8.14–10.94], FET $p = 0.0184$) whereas for those with C*04:01, it appears a non-minority background and a more advanced age (≥65 years old) may be predisposing to ICU hospitalization (Fig. 2 and Table S4; OR = 14.25 [12.24–16.26], FET $p = 0.0053$ and OR = 9.66 [8.27–11.05], FET $p = 0.0188$, respectively).

## DISCUSSION

We have previously identified the DPA1*02:02 class II allele as being prevalent in two other and independent cohorts, with patients of undisclosed ethnic background, but hospitalized in Wuhan, China (*Warren & Birol, 2020a*). Of populations with reported allele frequencies and an appreciable sampling size ($\geq$100 individuals), only Hong Kong Chinese and Japanese have DPA1*02:02 allele frequencies (55.8% and 43.5%, respectively; *Gonzalez-Galarza et al., 2020*; *Allele Frequency Net Database, 2020*) above its observed frequency (29.0%) in the COVID-19 positive cohort analyzed herein. The frequency of this allele in other qualifying populations tends to be generally lower, including in South African (Worcester, 15.6%), Norwegians (14.0%), Mexico Chiapas Lacandon Mayans (6.7%), United Kingdom Europeans (4.3%) and Spain Navarre Basques (2.2%). We note that the ethnic background of the *Overmyer et al. (2020)* cohort is heterogeneous, and individuals of possible European ancestry represent 45.0% of the COVID-19 positive cohort and 37.9% (11/29) of its DPA1*02:02 subset. In contrast, Asians represent only a minority of the cohort (2.0%) and its DPA1*02:02 subset 3.4% (1/29). It is important to note that, in the Overmyer cohort, DPA1*02:02 is not statistically associated with increased risk of hospitalization. The significant prevalence of this allele in the COVID-19 positive *vs.* negative cohorts (FET $p = 0.0174$) across all individuals, but also when looking only at females (FET $p = 0.0326$) or individuals from a non-minority background (FET $p = 0.0491$), and not any other demographics, may prove an important disease marker, which would need to be validated with additional, larger datasets and in independent studies. We also point out that we do not know the relationships, if any, between the New York COVID-19 patients. This is an important consideration and a potential source of bias in genetic association studies worth noting.

There are reports of disease associations with DPA1*02:02, C*04:01 and A*11:01, but they are few. Of note, the association of all aforementioned alleles with narcolepsy (*Ollila et al., 2015*; *Tafti et al., 2016*) and a known trigger for this auto-immune disease includes upper-airway infections and influenza vaccinations (*Han et al., 2011*; *Han et al., 2013*; *Heier et al., 2013*; *Moss et al., 2013*; *Montplaisir et al., 2014*; *Wijnans et al., 2013*). Susceptibility to narcolepsy may in fact be an indirect effect of HLA class I and the HLA class II DP isotype in response to viral and bacterial infections, including from influenza and *streptococcus* (*Ollila et al., 2015*; *Han et al., 2011*; *Aran et al., 2009*; *Mahlios, De la Herrán-Arita & Mignot, 2013*). It has since been reported that HLA-A*11 may be a susceptibility allele to influenza A(H1N1)pdm09 infection in some populations (*Dutta et al., 2018*) while another report implicates HLA-I allele C*04:01 with high HIV viral loads (*Olvera et al., 2015*). Further, it was recently demonstrated that the major histocompatibility complex (MHC) class II DR, DQ and DP isotypes play a role in mediating the cross-species entry of bat influenza viruses *in vitro* in human/animal cell lines and in mice where engineered MHC-II deficiency made them resistant to upper-respiratory tract infections (*Karakus et al., 2019*). It is therefore not a stretch to envision an involvement from these HLA class II isotypes in controlling the cellular entry of a broader range of viral agents *in vivo*.

In a recent study examining HLA susceptibility based on SARS-CoV-2 derived peptide (epitope) binding strengths (*Nguyen et al., 2020*), the HLA-I allele A*11:01 was *in silico* predicted to bind a large number of SARS-CoV-2 derived peptides ($n = 750$) with varying affinity [IC50 range 4.95–498.19, median = 149.62, mean = 182.28], and has been experimentally validated to bind SARS-CoV-2 peptide GLMWLSYFV (Tables S4 and S7 in *Nguyen et al. (2020)*. In contrast, C*04:01 was only predicted *in silico* to bind six SARS-CoV-2 peptides and at higher IC50 ranges [167.65 –469.30, median = 291.06, mean = 299.01] (Table S7 in *Nguyen et al. (2020)* suggesting a more limited ability to present epitopes to T cells and mount an appropriate immune response.

There have been a number of reports published on HLA alleles—COVID-19 associations this past year, and on cohorts from many jurisdictions including China (*Wang et al., 2019*), Italy (*Correale et al., 2020*; *Novelli et al., 2020*; *Pisanti et al., 2020*) and the UK (*Poulton et al., 2020*) among others. *Wang et al. (2019)* compared the HLA allele frequencies between a cohort of 82 Chinese individuals and a control population of bone marrow donors previously studied by the same group. Novelli and co-workers (*Novelli et al., 2020*) HLA typed a cohort of 99 Italian COVID-19 patients, and associated the observed allele frequencies with the HLA types in a reference group of 1,017 Italian individuals also previously studied by the same group. *Correale et al. (2020)* and and *Pisanti et al. (2020)* followed a different strategy; these two independent studies leveraged population scale genomics data retrieved from the Italian Bone-Marrow Donors Registry and the National Civil Protection Department. They correlated background HLA allele frequency data with mortality and morbidity rates across Italy to reach at starkly different conclusions on which HLA alleles may play a role in disease etiology and progression. Disagreement between these two studies (also distinct from the results of the other Italian study by Novelli et al.) highlight the importance of large cohorts with matched samples to infer the patient HLA alleles with better statistical significance. *Poulton et al. (2020)* characterized the HLA types of 80 COVID-19 patients in the UK on waiting lists for transplantation, and analyzed observed allele frequencies in comparison to a cohort of 10,000 deceased organ donors and a separate cohort of 308 SARS-CoV-2-negative individuals also on waiting lists for transplantation, the latter representing a matched demographics for the COVID-19 patients in their cohort. Interestingly, this is the only study that had any overlap between the alleles they flagged and the lists published by other studies cited above. Not surprisingly, the alleles they listed do not intersect with the alleles identified and presented herein and the three alleles we published earlier on a very small group of only eight patients. It is nonetheless intriguing to find little to no HLA allele overlap between these reports, including with those associated with the 2003 SARS outbreak, a related respiratory disease caused by a similar coronavirus (*Ovsyannikova et al., 2020*; *Lin et al., 2003*; *Sanchez-Mazas, 2020*). This could be explained, at least partially, by geographical differences and varying population allele frequencies in those cohorts, relatively small cohort sizes (<100 patients), differences in experimentation setup and/or other factors, including comorbidity status, that may be acting independently of HLA. We note that, since the beginning of the COVID-19 pandemic several large-scale studies reported no link between HLA and COVID-19 (*COVID-19 Host Genetics Initiative, 2021*; *Severe Covid-19 GWAS Group et al., 2020*; *Ben Shachar et al., 2021*). It is worth noting that

the MHC region of the genome is highly polymorphic and gene-dense, and without a direct measure of HLA, genome wide association studies (GWAS) have limitations when it comes to reporting associations in this region (*Kennedy, Ozbek & Dorak, 2017*). For instance, in the GWAS report from the COVID-19 Host Genetics Initiative (*COVID-19 Host Genetics Initiative, 2021*), the HLA locus was excluded from their analysis due to the heterogeneity in effect size observed between studies. Yet, in two other separate large-scale reports where HLA was directly measured and with appreciable COVID-19 patient cohort sizes from Italian/Spanish ($n > 1,000$) and Israeli ($n > 72,000$) populations, the absence of associations including with disease severity, is noteworthy (*Severe Covid-19 GWAS Group et al., 2020*; *Ben Shachar et al., 2021*).

Since we first reported on our observations (*Warren & Birol, 2020b*), a few additional notable HLA association with COVID-19 reports became available (*Khor et al., 2021*; *Weiner et al., 2020*; *Langton et al., 2021*; *Augusto et al., 2021*). Of interest, mining the US marrow donor program data bank, Augusto and co-workers uncovered an association between HLA-B*15:01 and asymptomatic SARS-CoV-2 infections (*Augusto et al., 2021*), which is stronger when haplotyped with HLA-DRB1*04:01. Interestingly, the latter HLA class II allele was also found associated with less severe symptoms in a regional cohort from the North of England (*Langton et al., 2021*). In a study from Germany, Weiner and co-workers (*Weiner et al., 2020*) HLA typed 233 COVID-19 patients from Germany, Spain, and Switzerland and found HLA-I C*04:01 to be associated with an increased risk of intubation in COVID-19 patients, consistent with our findings. In their preprint, they also present a validation of their observations, using the Overmyer (*Overmyer et al., 2020*) cohort data analyzed herein. The report from Japan looked at the HLA types of 190 COVID-positive patients, of which 53 had developed severe COVID-19 disease (*Khor et al., 2021*), and in their cohort they found HLA-A*11:01:01:01 to be significantly associated with the severe form of the disease. Although we found A*11:01 to be only nominally significantly associated with COVID-19 severity in the New York cohort, perhaps due to the sample size and relatively low allele prevalence, it is clear that with the pandemic rapidly evolving, HLA-COVID-19 association patterns (reviewed in *Migliorini et al., 2021*) are starting to emerge and are being corroborated by independent studies.

## CONCLUSIONS

Here, we predict HLA-I and HLA-II alleles from publicly available COVID-19 patient blood RNA-seq samples and identified several putative biomarkers. We had also reported one of these biomarkers, DPA1*02:02, in an earlier study. We postulate that patients with the allele may have an increased susceptibility for COVID-19. Further, other alleles, such as C*04:01, may be prognostic indicators of poor outcome. However, although it is well established that patient HLA profiles play a significant role in the onset and progression of infectious diseases in general, we caution against drawing overreaching conclusions from regional, and often limited, observations as is the case here. We note that recently published studies associating HLA alleles and COVID-19, by and large, disagree in their findings, which is not necessarily unexpected given that HLA frequency varies in different

populations and that patient cohorts may or may not be representative samplings of those populations. We also note the absence of associations in larger population-centric cohorts. We expect future studies with larger heterogeneous cohort sizes will help bring a clearer picture and demystify the role of patient HLA profiles, if any, in COVID-19 susceptibility and disease outcomes.

## ACKNOWLEDGEMENTS

The content of this paper is solely the responsibility of the authors, and does not necessarily represent the official views of the National Institutes of Health or other funding organizations.

### Funding

This work was supported by Genome BC and Genome Canada (281ANV); and the National Institutes of Health (2R01HG007182-04A1). The funders had no role in study design, data collection and analysis, decision to publish, or preparation of the manuscript.

### Grant Disclosures

The following grant information was disclosed by the authors:
Genome BC and Genome Canada: 281ANV.
The National Institutes of Health: 2R01HG007182-04A1.

### Competing Interests

The authors declare there are no competing interests.

### Author Contributions

- René L. Warren conceived and designed the experiments, performed the experiments, analyzed the data, prepared figures and/or tables, authored or reviewed drafts of the paper, and approved the final draft.
- Inanc Birol conceived and designed the experiments, prepared figures and/or tables, authored or reviewed drafts of the paper, and approved the final draft.

### Data Availability

The paper is a meta-analysis of a public dataset. The RNA-seq datasets analysed are available in the ENA repository: PRJNA660067, SRX9033799–SRX9033924.

The associated clinical data are available in the GEO repository: GSE157103.

We used third party tools for analysis and these tools are already in the public domain:
- HLAminer (https://github.com/bcgsc/HLAminer)
- seq2HLA (https://github.com/TRON-Bioinformatics/seq2HLA)
- OptiType (https://github.com/FRED-2/OptiType).

## Supplemental Information

Supplemental information for this article can be found online at http://dx.doi.org/10.7717/peerj.12368#supplemental-information.

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
