# Peer review of "HLA alleles measured from COVID-19 patient transcriptomes reveal associations with disease prognosis in a New York cohort"

_PeerJ, doi:10.7717/peerj.12368_

## Round 0.1 · original submission · Major Revisions

You are requested to address the insightful comments and concerns kindly raised by the two reviewers, importantly addressing the following two points 1) including a review on reports of negative association between HLA and COVID-19 and 2) reviewing and validating the statistical analyses.

Reviewer 1 has requested that you cite specific references. You may add them you believe they are especially relevant. However, I do not expect you to include these citations, and if you do not include them, this will not influence my decision.

Reviewer 1 ·

Basic reporting

Quite difficult to interpret what was done and how, on which patients. I felt this really weakened the submission. While it is not the easiest subject material to report on, I found it almost impossible to interpret.
At least two pertinent references were not include - very recent references are included so it is not clear why these were omitted as they appear to link in v well with this current work:
HLA-B*15:01 is associated with asymptomatic SARS-CoV-2 infection Danillo G. Augusto, Tasneem Yusufali, Noah D. Peyser, Xochitl Butcher, Gregory M. Marcus, Jeffrey E. Olgin, Mark J. Pletcher, Martin Maiers, Jill A. HollenbachmedRxiv 2021.05.13.21257065; doi: https://doi.org/10.1101/2021.05.13.21257065
And
Langton, DJ, Bourke, SC, Lie, BA, et al. The influence of HLA genotype on the severity of COVID-19 infection. HLA. 2021; 98: 14– 22. https://doi.org/10.1111/tan.14284

Experimental design

I am unclear as to exact patient selection, comorbids etc. Whether this is all in the Overmeyer paper or not I am sure - it needs to be represented. Given the relatively small numbers of subjects the diagnoses and comorbids are going to be extremely important.
Definitely needs a review from a statistician with well defined, appropriate expertise. I routinely deal with HLA data and I found this very difficult to manage.

Admission to study was based on symptoms/clinical presentation – then the two groups were distinguished by a positive or negative COVID test? What was used to determine covid positivity? Am I interpetting this correctly?

Is there a background population control group?

Could the authors explain more clearly how the HLA imputations compare to gold standard next generation sequencing?

Validity of the findings

I am sceptical though open minded. Needs further statistical review.
Discussion needs to be greatly improved, needs to be more focused in my opinion.

·

Basic reporting

The article is written in unambiguous, professional English.
The literature references are given for a sufficient background of the study. Only lacking are the genetic studies reporting negative findings for an association between HLA and COVID-19, which are in this case the studies with the best statistical power.
The article is well structured, and the tables and figures in the main manuscript represent the main findings in an appropriate way.
The raw data comprises a publicly available dataset, which is properly cited. The supplementary figures and tables would need improvement. For more details see the line by line comments in the general comments section.

Experimental design

The paper presents original primary research about the genetic association of HLA in COVID-19 on a publicly available dataset.
The research question is well defined and relevant but already studied with inconsistent results by several articles as also discussed by the authors. Additional to the discussion points mentioned by the authors I would like to highlight that the majority of the so far published studies is underpowered and results might in at least some cases be based on genetical stratification due to improper sampling rather than on true signals. Unfortunately, the same might be true for this study. The studied dataset is a mixture of ethnicities and the controls also showed symptoms compatible with COVID-19 infection and therefore might be genetically biased from healthy individuals.
Therefore, at least a correction for principal components is highly recommended if there is genetic information outside the HLA Region available. A stratified analysis only on European samples was performed but considering the strong differences in allele frequencies even within single countries and the mixed ethnical background in the present cohort, I expect the correction to be insufficient. Further, the statistic is bend to a limit by reducing the n for a Bonferroni correction based on an allele frequency and correcting for the single loci.

Validity of the findings

Currently I see the main information in the discussion only, where the authors point towards the inconsistency in the so far published papers regarding this topic. I criticize that the introduction gives the impression that the association of HLA with COVID-19 is already well defined and throughout the paper none of the negative findings in this field is cited.
Regarding their own findings the authors bend the statistic for Bonferroni correction by reducing the number of tests by considering only high frequency alleles and studying each locus independently.
Anyway, the conclusions are well stated and discussed.

Additional comments

Line-by-Line Notes:
L.34: “it is established that certain HLA alleles are disease determinants” – So far, all studies I am aware of with a positive finding are clearly underpowered and do not reproduce each other, while the big studies fail to identify any association (e.g. Shachar et al., 2021 and Ellinghaus et al., 2020). You also discuss this point later in the discussion. There were high expectations in HLA analysis but so far, no convincing association has been found.
L.87: Citation 15 does is not related to COVID-19. Further, the big genetic studies showing negative results for the HLA region (even after additional analysis because of expectations in this region) are not listed.
L.90: What is the relevance of the method used in the last paper at this place?
L.116: Maybe it would be nice to include a short note here as well on how you defined the HLA allele in case of disagreement between the models.
L.120: The term “frequency” is misleading, as you are not using the commonly used definition of frequency.
L.133: How is the patient/allele abundance rank defined?
Notes on the Figures Tables and Supplementary Material
• Fig1/Fig2: n samples for each curve
• What is the meaning of age :y for sample SRX9033804 (Table S1/S2)?
• Why are there 103 COVID samples in the Table S1/S2 but only 100 in the manuscript?
• Why is there only one allele per patient for each class II locus (Table S2)?
• What is the line in the Supplementary Figure S1? A linear regression does not make sense as the x axis shows an index.
• A summary of the ethnicity is given in the main text but no information in the listed data.

---

## Round 0.2 · Minor Revisions

Thank you so much for revising the manuscript. Please address the points indicated by reviewer 2. You are also requested to further cover the literature on HLA association with COVID-19 as indicated in the first review.

·

Basic reporting

The authors clearly improved the manuscript compared to the initial submission. Some uncertainties regarding bias in the data and resulting analysis remain.


Single small suggestions for language improvement:
Line 96: "... we report on HLA alleles with potential analytic (DPA1*02:02) and prognostic (A*11:01, C*04:01) value ...": For me the meaning of analytic and prognostic was not clear in the beginning even if the caption of Figure 2 clarifies the meaning. Maybe more descriptive words like susceptibility and severity/hospitalization free days might improve the understanding.

Line 176-179: I needed to read the sentence several times to figure out what the meaning of the values is. Maybe it would be an option to either split the brackets with the corrected p values into two or to name the n with something like "rank" or "#alleles".

Line 223-227: I would suggest to change the order slightly:
We note that the ethnic background of the Overmyer et al [27] cohort is heterogeneous, and individuals of possible European ancestry represent 45.1% of the COVID-19 positive cohort and 37.9% (11/29) of its DPA1*02:02 subset. In contrast, Asians represent only a minority of the cohort (1.9%) and its DPA1*02:02 subset 3.4% (1/29).

Figure 2: "for each HLA-I and HLA-II alleles" -> allele should be singular

Figure S1: Sentence fragment: "... consistency across."

Experimental design

Line 224: Why are the percentages in the COVID-19 positive cohort (n=100) no full percentages?

Table S1: I think it is suspicious that the only two COVID-19 positive Asian samples have adjacent sample IDs and similar age. This might be a hint for a bias in the dataset as both individuals might be of the same household or social network. Therefore the question if it is sure that no relatives are included in the dataset as this is a very strong bias for all genetic associations that should not be ignored. I would at least add a note to the discussion in case you cannot make sure that no relatives are included in the cohort. Even if no genetic association is noted as the COVID-19 infection spread mainly in clusters this is another (less important) bias as social networks are to some degree correlated to genetic background.

Table S2: Column names A1 / A2 confusing for people not in the HLA nomenclature esp. in comparison to Table S3.

Table 1/2: It would be great if you could also add the corrected p values for a better overview.

Validity of the findings

The points regarding weaknesses made after the first review were well addressed. But I recognised that no information was given about possible relatives in the dataset (see also 2. Experimental design)

---

## Round 0.3 · accepted · Accept

Thanks to the authors for adequately and satisfactorily addressing the reviewers’ comments. The work provides a comprehensive analysis of the association between HLA and COVID-19 in New York. The manuscript is ready for publication.

Please accept my sincere apologies for the delay in submitting the decision.

Reviewer 1 ·

Basic reporting

NA

Experimental design

NA

Validity of the findings

NA

Additional comments

I have recently suffered a family bereavement. I have read through this study as quickly as possibly as, despite me writing to Peer J to explain my circumstances I have been sent this again for review.
The authors have failed again to carry out an adequate review of the literature.
They have treated low resolution HLA typing as equivalent to high resolution studies and they have again interpreted the findings of references 23 and 24 incorrectly. If they were to re review the raw data presented in these two studies they would find agreement with their own study.